# First-Line Maintenance Treatment in Metastatic Colorectal Cancer (mCRC): Quality and Clinical Benefit Overview

**DOI:** 10.3390/jcm10030470

**Published:** 2021-01-26

**Authors:** Marta Martín-Richard, Maria Tobeña

**Affiliations:** 1Medical Oncology Department, Sant Pau Hospital, 08043 Barcelona, Spain; 2Medical Oncology Department, Quironsalud Hospital, 50006 Zaragoza, Spain; mtobenap@gmail.com

**Keywords:** maintenance treatment, sequential, continuous, intermittent treatment

## Abstract

Different strategies of maintenance therapy (sequential CT, intermittent CT, intermittent CT and MAbs, or de-escalation MAbs monotherapy) after first-line treatment are undertaken. Many randomized clinical trials (RCT), which evaluated these approaches, suffer from incorrect design, heterogenous primary endpoints, inadequate size, and other methodology flaws. Drawing any conclusions becomes challenging and recommendations are mainly vague. We evaluated those studies from another perspective, focusing on the design quality and the clinical benefit measure with a more objective and accurate methodology. These data allowed a clearer and more exact overview of the statement in maintenance treatment.

## 1. Introduction

The introduction of new drugs (Oxaliplatin and Irinotecan) used in doublets with 5 Fluorouracil (FU) added or not to monoclonal antibodies (MAbs) in first line has completely modified the survival rate of metastatic colorectal cancer (mCRC) patients but has increased the toxicity. Different strategies have been evaluated as maintenance treatments to improve tolerance without impairing outcome.

Several randomized clinical trials (RCT) evaluated these strategies. Different primary endpoints were chosen, and different designs were used. The main studies were planned to demonstrate a superiority from one arm in front of the other. However, when the purpose of a study is to show less toxicity or better tolerability, without impairment or a decrease in efficacity, a non-inferiority hypothesis is more suitable. Between them, the studies with an Overall Survival (OS) endpoint are the most critical in order to dismiss any real impairment. Other endpoints (Time to Progression (TTP), Duration Disease Control (DDC), or Tumor Control Disease (TCD)) are more debatable because there are not always surrogate markers of survival in mCRC. This flaw is amplified when the authors of a guideline want to settle recommendations. Firstly, they need to evaluate the available evidence. Several systems exist such as the Oxford Centre [1] or the GRADE [2]. Both are based on the category of the studies (randomized controlled study, controlled study without randomization, etc.) and use hazy and subjective definitions of the different categories. These systems do not heed size, bias existence, confounding issues, or other important issues.

In this context, we aimed to analyze those randomized studies that evaluate the different strategies, focusing first on design quality.

Secondly, we focused on clinical benefit evaluation. With this purpose, we assessed the studies according to the ESMO-Magnitude of Clinical Benefit Scale (ESMO-MCBS). ESMO-MCBS is a validated tool developed by the European Society of Medical Oncology (ESMO, Lugano, Switzerland) to evaluate the magnitude of clinical benefit for new anti-cancer drugs, the first version of which was published in May 2015) [3] and updated in 2017 (version1.1) [4]. This last version adds an evaluation, form 2c, for therapies that are not likely to be curative with a primary endpoint other than overall survival (OS), progression-free survival (PFS), or equivalent studies and was used for non-inferiority design studies. However, this evaluation form is based on overall response (OR) and improvement in some symptoms or quality of life (QoL). Frequently, studies with survival as primary endpoint do not report OR and commonly lack quality of life evaluation. Furthermore, they do not establish the cutoff for loss of OS that should be considered clinically admissible. As a result, this score seemed unusable and we proposed a score modification based on a Hazard ratio (HR) limit of 1.15, which means we considered an increase in the hazard of death of 15% or less acceptable [5,6]. In this way, we expected to provide more details, hoping to carry out useful and concise conclusions.

## 2. Material and Methods

Published randomized studies evaluating different strategies of maintenance treatment after first-line chemotherapy +- MAbs in advanced colorectal cancer were selected and reviewed. At the time of review, published randomized studies with MAbs evaluated only Cetuximab, Panitumumab, and Bevacizumab. In order to simplify the analysis, each study was included in one of the four categories, which correspond to one of these strategies: (1) sequential chemotherapy (CT) vs. upfront CT doublets, (2) continuous vs. intermittent CT doublets, (3) continuous doublets plus MAbs vs. intermittent, and (4) continuous CT plus MAbs vs. continuous MAbs monotherapy. All the studies in each category were evaluated putting the emphasis on the quality of trial design and the clinical benefit of the ESMO-MCBS score.

### 2.1. Quality of Trial Design (QTD)

The score used to evaluate the QTD includes three points: (1) achieved prespecified objective, (2) no change in predefined sample size and primary endpoint, and (3) adequate control arm. The level of evidence based on quality of the design ranged from 0 to 3 and, as a result, were categorized as low quality (0 to 1 point) or high quality (2 to 3 points) (Table 1).

### 2.2. ESMO-MCBS

Every study was evaluated for clinical benefit according to the ESMO-MCBS version 1.1. In studies with a superiority trial design, the evaluation form used was 2a (Table 2) or 2b (Table 3), depending on the primary endpoint (OS or PFS, respectively). In studies with a non-inferior trial design, the ESMO form 2c was used (Table 4).

ESMO non-inferiority (NI) modified: There are no set methods for determining limits in defining non-inferiority. We proposed an effect retention method to set the superior upper limit of 1.15. This method is supported by a significant systematic review of non-inferiority studies, as well as Federal Drug Administration (FDA) evaluation requirements [6]. The two other chosen points, loss of < 2.5 months in Median Survival (MS) and 3-year OS loss < 5%, seemed acceptable margins for clinical outcomes. This score was used for the evaluation of studies with non-inferiority design (Table 5).

## 3. Results

### 3.1. Sequential Treatment with Monotherapy vs. (VS) Upfront Chemotherapy (CT) Doublets Treatment

Five randomized studies of differing sizes evaluated this approach [7,8,9,10,11]. Similar patient characteristics and stratifying factors were used in these studies (Table 6).

#### 3.1.1. QTD

It is noteworthy to claim that the survival results expressed in the four former studies [7,8,9,10] were evaluated with the intention to treat a population. In the FOCCUS study [7] three strategies were compared. In planning to demonstrate an increase in OS from the second or third arm, in comparison with the first arm, a new primary endpoint of non-inferiority between the last two arms was added during the enrollment and was reached. That generated a global quality score of 2. The CAIRO trial [8] and the Cunningham study [10] had similar two-arm designs, but no differences were observed in the primary endpoint, OS. So, it generated a global quality score of 2. The other two studies shared a weak quality. The FFCD study [9] was prematurely closed and randomized only 410 patients from the 700 planned, as well as having an awkward primary endpoint. The more recent AIO-KKO study [11] adopted a non-inferiority design. Slow recruitment caused a sample size reduction and failed to demonstrate non-inferiority.

#### 3.1.2. ESMO-MCBS

Only one of the three studies with OS as primary endpoint can be evaluated with ESMO-MCBS evaluation form 2a, and its magnitude of clinical benefit was low. The FFCD study [9] was the only one with PFS as a primary endpoint, and was evaluated with ESMO-MCBS evaluation form 2b (PFS), with a low magnitude of clinical benefit.

#### 3.1.3. Non-Inferiority Evaluation

The FOCCUS and AIO KRK0110 studies were evaluable with ESMO-MCBS evaluation form 2c.

In the FOCCUS study the ESMO score was 0 as a result of the lack of an overall response report or improvement on QoL. On the contrary, our ESMO NI modified achieved a score of 2. In the AIO-KKO study the ESMO score was 0, and in the evaluation of OS in the ESMO NI Modified a score of 2 was obtained.

#### 3.1.4. Recommendations

In patients with a good PS and with non-curative intention, we could suggest the use of sequential treatment strategy. Strong evidence supports the lack of detrimental survival results in comparison to starting with doublets.

### 3.2. Continuous vs. Intermittent CT Treatment

Only three randomized studies have evaluated this intriguing question (Table 7).

#### 3.2.1. QTD

The OPTIMOX2 study [12] achieved the prespecified primary endpoint, higher-duration disease control (DDC) in the maintenance arm. However, the selection of this flawed endpoint, the premature closure of the study with only one third of the planned patients included, and the imbalance of patients submitted to surgery between the two arms favoring the maintenance arm (15% vs. 8%) weakened the quality of the study.

The COIN study [13], with a non-superiority design, was planned to show no OS differences in HR with a 1.62 threshold. Despite a large sample being reached, non-superiority could not be confirmed and neither non-inferiority between continuous nor intermittent treatment could be demonstrated. The Chinese study [14] was undertaken to show an increase in PFS between Capecitabine maintenance treatment vs. control that was achieved. The study got the highest score of 3.

#### 3.2.2. ESMO-MCBS

The primary endpoint in OPTIMOX2 study [12] was DDC endpoint and was not an evaluable one. If the study is evaluated with the ESMO-MCBS evaluation form 2b, the score is 3, as in the evaluation of the Chinese study, in favor of the maintenance arm. However, the real benefit is difficult to ascertain when no difference in OS is observed.

#### 3.2.3. Non-Inferiority Evaluation

In the COIN study, the improvement in several factors in QoL set a grade 3 in the ESMO-MCBS evaluation form 2c. In the evaluation of ESMO-NI modified, the low numerical differences in OS supported non-inferiority of the intermittent treatment with a high grade 3 score.

#### 3.2.4. Recommendations

Considering these results, intermittent treatment is highly recommended.

### 3.3. Continuous Doublets plus MAbs vs. Intermittent

Five studies evaluated planned de-escalation as a treatment strategy for patients without progression, after induction treatment with chemotherapy plus MAbs (Table 8).

#### 3.3.1. QTD

The AIO 0207 [15] and CAIRO3 [16] trials achieved their prespecified objectives and shared the highest quality design [3]. Conversely, SAKK 4106 [17] and PRODIGE 9 [18] did not achieve their prespecified objectives, and COIN-B [19] had to change its originally preplanned inclusion criteria when KRAS mutations were identified as predictors of resistance to EGFR MAbs and had the lowest score.

#### 3.3.2. ESMO-MCBS

The CAIRO3 trial [16] achieved its goal and, in evaluation with the ESMO-MCBS evaluation form 2b, had a score of 3. However, PRODIGE9 and COIN-B could not be evaluated with the ESMO-MCBS evaluation form 2b. In the former, the chosen TCD endpoint did not fulfil the ESMO evaluation score. On the other hand, COIN-B was designed as an exploratory study to complement the COIN-B trial. So, no conclusions could be drawn about the role of Cetuximab maintenance.

#### 3.3.3. Non-Inferiority Evaluation

The AIO 0207 and SAKK 4106 had non-inferiority designs but neither could be evaluated with the ESMO-MCBS evaluation form 2c. As in the first study, no significant differences were noted between arms in the mean value of general health status, the QoL score, or toxicity was reached. In the second one, QoL and toxicity were not evaluated. However, with the ESMO-NI modified the scores were 2 and 3, respectively.

#### 3.3.4. Recommendations

If a maintenance treatment is considered following first-line treatment with FOLFOX-Bevacizumab, a Bevacizumab–Fluoropyrimidine combination is recommended.

### 3.4. Continuous Doublets plus MAbs vs. Continuous Monotherapy Plus MAbs

Five randomized studies evaluated this new approach [20,21,22,23,24] (Table 9).

#### 3.4.1. QTD

All the studies shared high-quality design with the exception of MACBETH, which had the lowest score [1]. Not only were the inclusion/exclusion criteria modified during the study, excluding RAS- and BRAF-mutated tumors, but the endpoint was not achieved. The study did not have enough statistical power to detect the differences between the two arms. It is worth noting that in none of the studies was a favorable correlation rate between PFs and OS HRs shown.

#### 3.4.2. ESMO MCBS

Of the five studies evaluating this approach, only the VALENTINO2 study could be evaluated with the ESMO-MCBS evaluation form 2b. In the SAPPHIRE [24] study, nondefinitive screening comparisons were undertaken and, like the MACBETH study, were not evaluable.

#### 3.4.3. Non-Inferiority Evaluation

The MACRO 2 [21] study showed that PFS with Cetuximab maintenance was non-inferior to continuous treatment. However, the lack of improvement in tolerability, or the differences observed in survival, translated into a score of 0 in the ESMO evaluation form 2c, as well as in the ESMO-NI modified.

The maintenance of Bevacizumab was analyzed in a specifically non-inferiority designed trial, MACRO [20]. However, it could not be proven to be non-inferior to continuous treatment. The HR superior limit of 1.35 was observed, exceeding the 1.32 threshold considered. This result does not mean that continuous treatment is superior; it was simply not informative and was graded as 0 in the two evaluation scores.

#### 3.4.4. Recommendations

Regarding the EGFR inhibitors’ overview, Cetuximab maintenance was not shown to be superior to the control arm and, even if Panitumumab-5FU seemed to increase the PFS over Panitumumab, no improvement in OS was observed and no clinical benefit obtained. Therefore, no recommendation could be made.

## 4. Discussion

This detailed overview of the substantial RCT evaluating first-line maintenance in mCRC revealed the light and more common shadows on this landscape. Drawing conclusions, despite more than 18 studies appraising this setting, was not obvious and seemed lax. One of the principal reasons for this is the choice of an incorrect hypothesis. When maintenance treatment was evaluated, the endpoint was not to find an improvement in efficacy but to demonstrate less toxicity, better treatment tolerance, or improvement in quality of life. A non-inferiority design must be undertaken. Only seven of 18 studies followed this logic. Collecting this information is important in order to understand the real weight of the data. However, other guidelines (ESMO, French Intergroup, Australian Cancer Council) [25,26,27] do not reflect these facts. They are based on grading the evidence in trial categories (randomized, quasi-randomized, observational), adjusted by confusing definitions, as “further research is very unlikely to change the confidence or to have an impact in confidence of the effect”. In this way, in the ESMO guidelines, three of the four recommendations in maintenance treatment grade IV-A were equal to expert opinion. In the French Intergroup Clinical guideline, only two statements related to maintenance therapy were exposed and one of them was grade C, meaning that ”further investigation is likely to have an important impact in our confidence of the effect”. Therefore, the main guidelines lacked focus on maintenance treatment issues and contributed only weak information.

A further step to establish a recommendation must be to evaluate the clinical benefit. For this purpose, in the maintenance treatment setting, it is not only important to evaluate the decrease in toxicity but, even more so, the improvement in some symptoms or in quality of life. It should be a requirement to establish which limit of impairment in the risk of death is clinically acceptable. ESMO-MCBS only evaluated the first two points but did not rely on any objective measure for this purpose.

In our proposal, we selected clinically acceptable margins of the HR 1.15. The margin selected was supported by the effect retention method. This methodology of selecting the non-inferiority margin is the least criticized by experts, endorsed by larger systematic revisions [5], and used, as well as being required, by the FDA for new drug approval based on non-inferiority studies [6]. This HR must be criticized and, from our perspective, must be taken only as a proof of concept.

Furthermore, most of the studies with an OS primary endpoint did not detail OR (two of four). In studies where sequential treatment was compared to an upfront CT, the selection of which OR (after first line or second line) must be used for comparison were highly questionable and dismissed the ESMO evaluation. QoL was barely evaluated (five of 18) and, if it was assessed, a small number of patients fulfilled the QoL test, resulting in poor representative results. If we take as an example the FOCCUS study, where an amendment was undertaken to include a non-inferiority design comparison between the FU and sequential doublet arm vs. upfront doublet treatment arm, the primary endpoint was achieved. Nevertheless, no OR or QoL was reported and the study’s ESMO-MCBS-derived score of 0 was non-informative. With our proposal, a score of 2 was observed, eliciting more useful information. Similarity was noted in the SAKK study evaluation. Non-inferiority in TTP with control vs. Beva maintenance was observed. However, the ESMO-MCBS score was 0, whereas a high score was obtained in our proposed scale.

The other guidelines (ESMO, French Intergroup, American Society of Clinical Oncology ASCO resources stratified guidance, and Cancer Council Australian) [25,26,27,28] and our review agreed that, after a first line of FOLFOX and bevacizumab, to continue with Beva and Fluoropyrimidine is the most highly recommended. ESMO and the Australian Cancer Council believe that there is not enough data to define the best maintenance treatment after a first line with CT and EGFR inhibitors. Parameters that identified subgroups of patients that could benefit from more, or less, active maintenance strategies were lacking and seemed to be of great interest. It is regrettable that the inclusion of more than 3300 patients did not provide evidence of this clinical need.

Altogether, even if no new results from methodologically well-planned studies are expected in the near future, refining the clinical benefit evaluation tools would seem to be the most correct way to move forward.

## 5. Conclusions

Several studies were established to evaluate different approaches in first-line maintenance treatment in mCRC. Several issues such as different chosen endpoints and inaccurate design studies, among others, made it difficult to draw clear conclusions.

Sequential treatment does not seem to be detrimental in comparison to starting with combined CT. If a doublet CT is started as up-front therapy, intermittent treatment does not seem to compromise the outcome, improves the quality of life, and is to be recommended. Unfortunately, with the use of MAbs, no conclusion was reached with regard to decreasing toxicity without jeopardizing outcomes. Concerns about the methodology used in the studies’ design and the lack of accurate evaluation tools emerged as hurdles in arriving at conclusions. A huge effort to solve them would be a very useful step in making progress.

## Figures and Tables

**Table 1 jcm-10-00470-t001:** Items for quality trial design (QTD) analysis.

Variables to Analyze	Yes	No
Change in the originally pre-planned, sample size or primary end-point	0	1
Achieved pre-specified objective	1	0
Adequate control arm ^1^	1	0

^1^ At the time of trial design. Level of evidence based on the quality of the design: 0 to 1, low quality; 2 to 3, high quality.

**Table 2 jcm-10-00470-t002:** **ESMO-MCBS v1.1 evaluation form 2A**. For therapies that are not likely to be curative with primary endpoint of OS.

If Median OS with the Standard Treatment > 12 Months ≤ 24 Months	If Median OS with the Standard Treatment is ≤ 12 Months	If Median OS with the Standard Treatment > 24 Months
**GRADE 4:**	**GRADE 4:**	**GRADE 4:**
• HR ≤ 0.70 AND gain ≥ 5 months	• HR ≤ 0.65 AND gain ≥ 3 months	•HR ≤ 0.70 AND gain ≥ 9 months
• Increase in 3 years survival alone ≥10%	• Increase in 2 years survival ≥ 10%	•Increase in 5 years survival alone ≥ 10%
**GRADE 3:**	**GRADE 3:**	**GRADE 3:**
HR ≤ 0.70 AND gain ≥ 3–<5 months	HR ≤ 0.65 AND gain ≥ 2.0–<3 months	HR ≤ 0.70 AND gain ≥ 6–<9 months
**GRADE 2:**	**GRADE 2:**	**GRADE 2:**
• HR ≤ 0.70 AND gain ≥ 1.5–<3 months	• HR ≤ 0.65 AND gain ≥ 1.5–<2.0	•HR ≤ 0.70 AND gain ≥ 4–<6 months
• HR > 0.70–0.75 AND gain ≥ 1.5 months	• HR > 0.65–0.70 AND gain ≥ 1.5 months	•HR > 0.70–0.75 AND gain ≥ 4 months
**GRADE 1:**	**GRADE 1:**	**GRADE 1:**
HR > 0.75 OR gain < 1.5 months	HR > 0.70 OR gain < 1.5 months	HR > 0.75 OR gain < 4 months
Non-curative setting grading—5 and 4 (substantial benefit), 3 (moderate benefit), 2 and 1 (negligible benefit)	Non-curative setting grading—5 and 4 indicates a substantial magnitude of clinical benefit	Non-curative setting grading—5 and 4 indicates a substantial magnitude of clinical benefit

PFS: progression-free survival, OS: overall survival, HR: hazard ratio.

**Table 3 jcm-10-00470-t003:** **ESMO-MCBS v1.1 evaluation form 2B.** For therapies that are not likely to be curative with primary endpoint of PFS.

If Median PFS with Standard Treatment ≤ 6 Months	If Median PFS with Standard Treatment > 6 Months
**GRADE 3:** HR ≤ 0.65 AND gain ≥ 1.5 months **GRADE 2:** HR ≤ 0.65 BUT gain < 1.5 months **GRADE 1:** HR > 0.65 Non-curative setting grading—5 and 4 indicates a substantial magnitude of clinical benefit	**GRADE 3:** HR ≤ 0.65 AND gain ≥ 3 months **GRADE 2:** HR ≤ 0.65 BUT gain < 3 months **GRADE 1:** HR > 0.65 Non-curative setting grading—5 and 4 indicates a substantial magnitude of clinical benefit
**Early stopping or crossover:** • Did the study have an early stopping rule based on interim analysis of survival? • Was the randomization terminated early based on the detection of overall survival advantage at interim analysis? If the answer to both is “yes”, then see letter “E” in the adjustment section below	**Early stopping or crossover:** • Did the study have an early stopping rule based on interim analysis of survival? • Was the randomization terminated early based on the detection of overall survival advantage at interim analysis? If the answer to both is “yes”, then see letter “E” in the adjustment section below
**Toxicity assessment** Is the new treatment associated with a statistically significant incremental rate of: «Toxic» death > 2%, Cardiovascular ischemia > 2%, Hospitalization for «toxicity» > 10%, Excess rate of severe congestive heart failure > 4%, Grade 3 neurotoxicity > 10%, Severe other irreversible or long lasting toxicity > 2% (Incremental rate refers to the comparison versus standard therapy in the control arm)	**Toxicity assessment** Is the new treatment associated with a statistically significant incremental rate of: «Toxic» death > 2%, Cardiovascular ischemia > 2%, Hospitalization for «toxicity» > 10%, Excess rate of severe congestive heart failure > 4%, Grade 3 neurotoxicity > 10%, Severe other irreversible or long lasting toxicity > 2% (Incremental rate refers to the comparison versus standard therapy in the control arm)
**Quality of life/Grade 3–4 toxicities assessment** • Was QoL evaluated as secondary outcome? • Does secondary endpoint QoL show improvement? Are there statistically significantly less grade 3–4 toxicities impacting on daily well-being? (This does not include alopecia, myelosuppression, but rather chronic nausea, diarrhea, fatigue, etc.)	**Quality of life/Grade 3–4 toxicities assessment** • Was QoL evaluated as secondary outcome? • Does secondary endpoint QoL show improvement? Are there statistically significantly less grade 3–4 toxicities impacting on daily well-being? (This does not include alopecia, myelosuppression, but rather chronic nausea, diarrhea, fatigue, etc.)
**Adjustments** A: When OS as secondary endpoint shows improvement, it will prevail and the new scoring will be done according to form 2a. B: Downgrade 1 level if there is one or more of the above incremental toxicities associated with the new medicine. C: Downgrade 1 level if the medicine ONLY leads to improved PFS (mature data shows no OS advantage) and QoL assessment does not demonstrate improved QoL D: Upgrade 1 level if improved QoL or if less grade 3–4 toxicities that bother patients are demonstrated. E: Upgrade 1 level if study had early crossover because of early stopping or crossover based on detection of survival advantage at interim analysis. F: Upgrade 1 level if there is a long-term plateau in the PFS curve, and there is > 10% improvement in PFS at 1 year. Highest magnitude clinic benefit grade that can be achieved grade 4. Non-curative setting grading—5 and 4 indicates a substantial magnitude of clinical benefit	**Adjustments** A: When OS as secondary endpoint shows improvement, it will prevail and the new scoring will be done according to form 2a. B: Downgrade 1 level if there is one or more of the above incremental toxicities associated with the new medicine. C: Downgrade 1 level if the medicine ONLY leads to improved PFS (mature data shows no OS advantage) and QoL assessment does not demonstrate improved QoL D: Upgrade 1 level if improved QoL or if less grade 3–4 toxicities that bother patients are demonstrated. E: Upgrade 1 level if study had early crossover because of early stopping or crossover based on detection of survival advantage at interim analysis. F: Upgrade 1 level if there is a long-term plateau in the PFS curve, and there is > 10% improvement in PFS at 1 year. Highest magnitude clinic benefit grade that can be achieved grade 4. Non-curative setting grading—5 and 4 indicates a substantial magnitude of clinical benefit.

PFS: progression-free survival, OS: overall survival, QoL: quality of life.

**Table 4 jcm-10-00470-t004:** ESMO-MCBS v1.1 evaluation form 2c. For therapies that are not likely to be curative with primary endpoint other than OS, PFS, or equivalence studies.

**Primary Outcome is Toxicity or Quality of Life AND Non-Inferiority Studies**
**GRADE 4:** Reduced toxicity or improved QoL (using a validated scale) with evidence for statistical non-inferiority or superiority in PFS/OS
**GRADE 3:** Improvement in some symptoms (using a validated scale) BUT without evidence of improved overall QoL
**Primary Outcome is Response Rate**
**GRADE 2:** RR is increased ≥ 20% but no improvement in toxicity/QoL/PFS/OS
**GRADE 1:** RR is increased < 20% but no improvement in toxicity/QoL/PFS/OS
**Final magnitude of clinical benefit grade 1–4** Non-curative setting grading—5 and 4 indicates a substantial magnitude of clinical benefit

PFS: progression-free survival, OS: overall survival, QoL: quality of life, RR: response rate.

**Table 5 jcm-10-00470-t005:** ESMO-M non inferiority (NI)designed studies modified.

	Yes	No
Lost in median survival < 2.5 months	1	0
Lost 3 year-OS < 5%	1	0
HR < 1.15	1	0

OS: overall survival, HR: hazard ratio. Recommendation grades: low 0 to 1, strong 2 to 3.

**Table 6 jcm-10-00470-t006:** Sequential treatment with monotherapy versus (vs.) upfront chemotherapy (CT) doublets treatment.

Trial	N	Treatment Arms	Primary Endpoint	PFS	OS	QTD	ESMO-MCBS Form 2a (OS)	ESMO-MCBS Form 2b (PFS)	ESMO-MCBS Form 2c (NI)	ESMO-NI Modified
**FOCCUS** **Seymour MT. Lancet 2007**	2135	A:FU and Irinotecan B: FU and doublets C:Doublets	OS A vs. B/C OS B vs. C (HR 1.18)	NR	13.9 HR AvsB 0.94 (0.84–1.05) 15 15.2 HR A vs. C 0.88 (0.79–0.98) *p* 0.02 HR C vs. B 1.06 (0.97–1.17)	2	NA	NA	0	2
**CAIRO** **Koopma M. Lancet 2007**	820	Capecitabine and Irinotecan and CAPOX vs. CAPOX or CAPIRI	OS (HR 0.80)	5.8 vs. 7.8 m HR 0.77 (0.67–0.98) *p* 0.002	16.3 vs. 17.4 m HR0.92 (0.79–1.08) *p* 0.32	2	NA	1	NA	NA
**FFCD** **Ducruex M. Lancet Oncol 2011**	410	FU and Doublets vs. Doublets	PFS	10.9 vs. 10.3 m HR 0.95 (0.77–1.16) *p* 0.61	16.4 vs. 16.2 m HR1.02 (0.82–1.27) *p* 0.85	1	NA	NA	NA	NA
**XELANTIRI. Cunningham D. 2009**	725	FU–FOLFOX-FOLFIRI	OS	7.9 vs. 5.9 m HR 0.67 (0.58–0.79) *p* < 0.0001	15.9 vs. 15.2 m HR 0.93 (0.78–1.10)	2	1	3	NA	NA
**AIO KRK0110** **Modest DP. J Clin Oncol 2018**	421	FU-Bevacizumab followed by Irinotecan vs. Fluoropyrimidine-Irinotecan-Bevacizumab	TFS	NR	21.9 vs.	1	NA	NA	0	2
					23.5 m					
				NR	HR 0.84					
				HR 0.70 *p* < 0.001	(0.66–1.06) *p* 0.14					

PFS: progression-free survival, OS: overall survival, QTD: quality of trial design, NI: non-inferiority, TFS: time to failure of strategy, NR: not reported, m: months, NA: not applicable, non-inferiority studies (bottom in blue).

**Table 7 jcm-10-00470-t007:** Continuous vs. intermittent CT treatment.

Trial	N	Treatment Arms	Primary Endpoint	PFS	OS	QTD	ESMO-MCBS Form 2a (OS)	ESMO-MCBS Form 2b (PFS)	ESMO-MCBS Form 2c (NI)	ESMO-NI Modified
**OPTIMOX 2.** **Chibaudel B. J Clin Oncol 2009**	202	FOLFOX × 6 cycles vs. FOLFOX followed by FU/LV	DDC	8.6 vs. 6.6 m HR 0.61(NR) *p* 0.001	23.8 vs. 19.5 HR 0.88 (NR) *p* 0.42	2	NA	3	NA	NA
**COIN.** **Adams RA. Lancet 2011**	1630	FOLFOX or CAPOX ± CETUXIMAB × 3 vs. FOLFOX or CAPOX ± CETUXIMAB	OS (HR < 1.62)	NR	15.8 vs. 14.4 m HR 1.084 (1.008–1.65) *p* NR	2	NA	NA	3.	3
**Luo HY, Ann Oncol 2016**	274	CAPOX/FOLFOX × 18weeks vs. CAPOX/FOLFOX × 18weeks followed by CAPECITABINE	PFS	6.4 vs. 3.4 m HR 0.54 (0.42–0.7) *p* <0.001	25.6 vs. 23.3 m HR 0.85 (0.64–1.11) *p* 0.22	3	NA	3	NA	NA

PFS: progression-free survival, OS: overall survival, QTD: quality of trial design, NI: non-inferiority, DDC: duration of disease control, m: months, NR: not reported, NA: not applicable, non-inferiority studies (bottom in blue).

**Table 8 jcm-10-00470-t008:** Treatment with continuous CT doublets plus monoclonal antibodies (MAbs) vs. intermittent treatment.

Trial	N	Treatment Arms	Primary Objective	PFS	OS	QTD	ESMO-MCBS Form 2a (OS)	ESMO-MCBS Form 2b (PFS)	ESMO-MCBS Form 2c (NI)	ESMO-NI Modified
**AIO 0207** **Hegewisch-Becker S. Lancet Oncol 2015**	472	BEVACIZUMAB-FU/LV vs. BEVACIZUMAB vs. NO treatment	TFS	6.3 vs. 4.6 vs. 3.5 m *p* < 0·0001	20.2 vs. 21.9 vs. 23.1 m *p* 0.77	3	NA	NA	0	2
				BEV vs. BEV FluoHR HR 1.34 (1.06–1.70) *p* 0.015					0	1
				NO vs. BEV Fluo HR 2.09 (1.64–2.67) *p* < 0.0001					0	2
				NO vs. BEV HR 1.45 (1.15–1.82) *p* 0.0018.						
**SAKK 4106** **Koeberle D, Ann Oncol 2015**	262	NO treatment vs. BEVACIZUMAB	TTP	9.5 vs. 8.5 m HR 0.75 (0.59–0.97) *p* 0.025	25.4 vs. 23.8 m HR 0.83 (0.63–1.1) *p* 0.2	2	NA	NA	0	3
**CAIRO 3** **Simkens L, Lancet 2015**	588	NO treatment vs. CAPECITABINE-BEVACIZUMAB	PFS2	8.5 vs. 11.7 m HR 0.67 (0.56–0.81) *p* < 0.0001	18.1 vs. 21.6 m HR 0.83 (0.68–1.01) *p* 0.06	3	NA	3	NA	NA
**PRODIGE 9** **Aparicio T. J Clin Oncol 2018**	491	NO treatment vs. BEVACIZUMAB	TCD	9.9 vs. 9.5 m HR 0.89 (0.70–1.13) *p* 0.33	27.6 vs. 28.5 m HR 1.11 (0.86–1.45) *p* 0.424	2	NA	NA	NA	NA
**COIN-B** **Wasan H. Lancet Oncol 2014**	169	NO treatment vs. CETUXIMAB	FFS at 10 months	3.1 vs. 5.8 m HR NR	16.8 vs. 22.2 m HR NR	2	NA	NA	NA	NA

PFS: progression-free survival, OS: overall survival, QTD: quality of trial design, NI: non-inferiority, TFS: time to failure of strategy, TTP: time to progression, TCD: tumor control duration, FFS: failure-free survival, m: months, NA: not applicable, BEV: Bevacizumab, Fluo: Fluoropyrimidine, No: no treatment, non-inferiority studies (background in blue).

**Table 9 jcm-10-00470-t009:** Continuous treatment with chemotherapy doublets plus MAbs vs. continuous treatment with monotherapy plus MAbs.

Trial	N	Treatment Arms	Primary Objective	PFS	OS	QTD	ESMO-MCBS Form 2a (OS)	ESMO-MCBS Form 2b (PFS)	ESMO-MCBS Form 2c (NI)	ESMO-NI Modified
**MACRO.** **Diaz-Rubio E. Oncologist 2012**	480	CAPOX-BEVACIZUMAB vs. CAPOX-BEVACIZUMAB followed by BEVACIZUMAB	PFS	10.4 vs. 9.7 m HR 1.10 (0.89–1.35) *p* 0.38 (predefined noninferiority limit: 1.32	23.2 vs. 20.0 m HR 1.05 (0.85–1.30) *p* 0.65	2	NA	NA	NA	0
**MACRO 2** **Aranda E Eur J Cancer 2018**	193	FOLFOX-CETUXIMAB × 8 and CETUXIMAB vs. FOLFOX-CETUXIMAB	PFS at 9 months	9 vs. 10 m HR 1.2 (0.8–1.8) *p* 0.39	23 vs. 27 m HR 1.2 (0.9–1.8) *p* = 0.2649	3	NA	NA	NA	0
**MACBETH** **Cremolini C. JAMA Oncol 2018**	143	FOLFOXIRI-CETUXIMAB vs. FOLFOXIRI-CETUXIMAB and then BEVACIZUMAB	10-month PFR	10.1 vs. 9.3 m HR 0.83 (0.57–1.21) *p* NR	33.2 vs. 32.2 m HR 0.92 (0.57–1.47) *p* NR	1	NA	NA	NA	NA
**VALENTINO2** **Pietrantonio F. JAMA Oncol 2019**	299	FOLFOX-PANITUMUMAB × 8 vs. FOLFOX plus PANITUMUMAB	10-month progression-free survival (PFS)	12 vs. 9.9 m HR 1.51 (1.11–2.07) *p* 0.009	NRD vs. NRD HR 1.13 (0.71–1.81) *p* 0.60	2	NA	3	NA	NA
**SAPPHIRE Munemoto. Eur J Cancer 2019**	277	FOLFOX-PANITUMUMAB × 6 continuous vs. FOLFOX-PANITUMUMAB × 6 and PANITUMUMAB	PFS rate at 9 months	9.1 vs. 9.3 m HR 0.93 (0.60–1.43)	NRD vs. NRD, HR 1.41 (0.69–2.88) *p* NR	2	NA	NA	NA	NA

PFS: progression-free survival, OS: overall survival, QTD: quality of trial design, NI: non-inferiority, PFR: progression-free rate, m: months, NA: not applicable, NRD: non reported; non-inferiority studies (background in blue).

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
