# Peer review of "First-Line Maintenance Treatment in Metastatic Colorectal Cancer (mCRC): Quality and Clinical Benefit Overview"

_jcm, 2021, doi:10.3390/jcm10030470_

Round 1

Reviewer 1 Report

It is an interesting paper but it can be significantly improved:

  • The abstract should be reviewed a re-written (to avoid mistakes)
  • The Introduction section lacks any bibliographic reference to current standard first-line treatment and methods of literature evaluation (ESMO benefit scale)
  • Title and Headings are telegraphic and should be completed: e.g., "first line maintenance" -> Maintenenance after first line therapy", or "sequential vs doublets" -> sequential treatment with monotherapy vs upfront CT doublets", etc.
  • Headings should be re-numered correctly
  • Table 1 is missing 

Author Response

You can find below a complete answer to all of your comments.

It is an interesting paper but it can be significantly improved:

  • The abstract should be reviewed and re-written (to avoid mistakes)

As requested, the abstract has been re-written and we expect all these issues to have been solved.

  • The Introduction section lacks any bibliographic reference to current standard first-line treatment and methods of literature evaluation (ESMO benefit scale)

The introduction has been completely modified. We have included more details about the evaluation scale used and bibliographic references have been added.

  • Title and Headings are telegraphic and should be completed: e.g., "first line maintenance" -> Maintenance after first line therapy", or "sequential vs doublets"

Titles and headings have been modified following your suggestion.

  • Headings should be re-numbered correctly

New tables have been included and headings modified.

  • Table 1 is missing 

Table1 was originally included as a supplementary document. Unfortunately, it seems it was not included in your package.

Reviewer 2 Report

This paper is of poor quality both in form and in substance

The authors don’t even refer to the tools they say they use to assess the quality of the tests

 Lack of diagrams or tables that would make it possible to see more clearly in everything they say

There are some misunderstandings

All this to arrive at conclusions that are already commonly accepted...

Author Response

You can find below a complete answer to all of your comments.

  • This paper is of poor quality both in form and in substance

We do not agree with this evaluation. In any case the paper has been totally reviewed and rewritten as an original article.  Aside from a thorough review of the literature with an accurate methodology, we propose recommendations that are contextualized with those from other authors. We hope this new revision will meet your standards.

  • The authors don’t even refer to the tools they say they use to assess the quality of the tests

In order to clarify the methodology used, we add all the information in the ‘Material and Methods’ section.

  • Lack of diagrams or tables that would make it possible to see more clearly in everything they say

Several tables have been added in order to give more light to the explanations.

  • There are some misunderstandings

We expect the new wording to clarify any misunderstandings.

  • All this to arrive at conclusions that are already commonly accepted...

An exhaustive analysis of the published literature was carried out.  This rigorous evaluation points out the highs and lows of the studies analyzed and allows us to elicit our recommendations. In the ‘Discussion’, we consider whether we consider if our recommendations correspond to other authors suggestions. Therefore, we try to give tips to allow others to do their own critical review 

Reviewer 3 Report

The introduction is unclear and the objectives of the review are not fully understood.
You used an Esmo-score without specifying its criteria or inserting a bibliographic reference.
The acronyms used for the first time are not always specified in full and make their interpretation difficult.
The studies chosen for the review are not all impactful in clinical practice, they are not explained even briefly or with a table, it is not clear how they are evaluated and how you come for each to formulate the paragraph "ESMO clinical benefit magnitude" and the following "Overall interpretation to grade recommendations"

The conclusions are unclear.

Author Response

You can find below a complete answer to all of your comments

  • The introduction is unclear and the objectives of the review are not fully understood.

The paper has been totally reviewed and rewritten as an original article. Besides a thorough review of the literature with an accurate methodology, we propose recommendations that are contextualized with those from other authors.

You used an Esmo-score without specifying its criteria or inserting a bibliographic reference

The introduction has been completely modified. We have included more details about the evaluation scale used and bibliographic references have been added.  Greater details are included in the ‘Material and Methods’ section.

  • The acronyms used for the first time are not always specified in full and make their interpretation difficult

Acronyms have been revised and modified.

  • The studies chosen for the review are not all impactful in clinical practice, they are not explained even briefly or with a table, it is not clear how they are evaluated and how you come for each to formulate the paragraph "ESMO clinical benefit magnitude" and the following "Overall interpretation to grade recommendations"

We expect to have explained those concerns in the ‘Material and Methods’

  • The conclusions are unclear

As previously mentioned, the paper has been completely re-written. We expect   the recommendations  to be more easily understood.  In the ‘Discussion’, we consider if our recommendations are in line with the other authors suggestions. Therefore, we try and give tips for their own critical review.